# The Mechanism by Which Umbrella-Shaped Ratchet Trichomes on the *Elaeagnus angustifolia* Leaf Surface Collect Water and Reflect Light

**DOI:** 10.3390/biology12071024

**Published:** 2023-07-20

**Authors:** Zhanlin Bei, Xin Zhang, Xingjun Tian

**Affiliations:** 1School of Life Sciences, Nanjing University, Nanjing 210023, China; realpal00147@163.com; 2School of Biological Science and Engineering, North Minzu University, Yinchuan 750021, China; x_zhang@nmu.edu.cn

**Keywords:** *Elaeagnus angustifolia* leaf, umbrella-shaped ratchet trichomes, water collection, light reflection

## Abstract

**Simple Summary:**

Plants in arid areas have evolved narrow linear leaves or degenerated into cone-shaped spines to reduce solar radiation and excessive water loss. We have demonstrated for the first time the widely distributed sub-tree species of *Elaeagnus angustifolia* in arid areas. *E. angustifolia* has large- and medium-sized leaves, and the surface of the leaves is covered with umbrella-shaped ratchet trichomes that can capture moisture in the air and reflect solar radiation. This study reveals that the trichomes on the *E. angustifolia* leaf surface can collect water and reflect light, which provides a reference for bionic development and research in atmospheric water harvesting, seawater desalination, energy management, microfluidic control, and daytime radiative cooling.

**Abstract:**

Leaves are essential for plants, enabling photosynthesis and transpiration. In arid regions, water availability limits plant growth. Some plants, like *Elaeagnus angustifolia*, a sandy sub-tree species widely distributed in arid and semi-arid regions, have unique leaf structures to reduce water loss and solar radiation. Here, we describe the leaves of *Elaeagnus angustifolia* L., with special functioning trichomes. Through leaf submicroscopic structure observation, in situ water collection experiments, photosynthesis measurements, and reflection spectrum analysis, we investigated *E. angustifolia* leaves, focusing on their functioning trichomes. These trichomes capture water vapor, reflect UV and NIR light, and possess a 3D interface structure composed of 1D and 2D structures. The 1D conical structure captures water droplets, which are then gathered by the radial conical structure and guided towards the stomata through wedge-shaped grooves on the 2D umbrella structure. The trichomes also reflect sunlight, with micropapillae reflecting UV light and the umbrella structure reflecting NIR light. These mechanisms reduce leaf temperature, respiration, and water transpiration, protecting against solar radiation damage. This study provides insights into water collection and light-reflection mechanisms, revealing adaptive strategies of plants with large leaves in arid regions.

## 1. Introduction

In arid regions characterized by clear skies and intense sunlight [1,2,3], efficient water acquisition, water conservation, and reduced solar radiation and heat absorption are critical for the survival of plants [4,5,6]. Plant leaves, like other organs, experience severe drought stress [7]. Consequently, leaves have developed an ingenious and effective mechanism to absorb water and reflect light in response to environmental stress. For instance, *Opuntia microdasys* leaves in desert areas possess specialized spines and glandular trichomes that minimize solar radiation and water loss during gas exchange and transpiration. Moreover, these clustered needle-like spines enable the directional collection of airborne water droplets [8], which sustains cactus survival in arid environments for several days. Another example is *Syntrichia caninervis*, a perennial plant that thrives in arid regions. Its leaves feature long white “awns” at their tips, which efficiently collect and transport water from the air, while also reflecting sunlight to minimize water evaporation. This adaptation allows *S. caninervis* to flourish in arid environments [9]. In semi-humid regions, the large palmate leaves of *Populus tomentosa* possess a non-wettability hollow white tomentose layer on their back surfaces, which reflects up to 55% of sunlight. This feature significantly reduces leaves’ heat absorption and shields them from intense solar radiation [10]. However, little is known about the drought-resistant strategies of plants with large- and medium-sized leaves in arid and semi-arid regions.

*Elaeagnus angustifolia* L. (Elaeagnaceae) is a deciduous tree or large multilayer shrub with small size and spines. It is widely distributed in arid and semi-arid areas [11]. This species is commonly known as “Russian olives” or “oleaster” [12] and is native to southern Europe, Central Asia, and the Western Himalayas [13]. During the early 20th century, it was introduced from Eurasia to Canada, the United States, the Mediterranean coast, Southern Russia, Iran, and India [14]. The fruit of *E. angustifolia* is highly nutritious, containing proteins, sugars, vitamins, and minerals, making it a valuable natural food and medicinal herb [15]. The climate conditions suitable for *E. angustifolia* are mainly found in cold and dry warm and mid-temperate zones in winter, as well as in arid and semi-arid regions between 30° N and 50° N in China [16]. *E. angustifolia* thrives in these areas, where the annual rainfall does not exceed 300 mm [17,18,19,20]. Due to its ability to withstand severe drought, high salinity, cold, and wind stress [21], *E. angustifolia* plays a crucial role in preserving ecosystem functionality in arid areas [22].

The leaves of *E. angustifolia* are medium-sized and lanceolate in shape. They possess trichomes on both the upper and lower surfaces, exhibiting a silver-white color [23,24]. Upper sun leaves are smaller, more slender, and thicker than the lower, half-exposed shade leaves [25]. The foliage of a single *E. angustifolia* plant displays a gradient from silver-white to dark green, with peltate trichomes on the upper sun-exposed leaves, pedestalled trichomes on the medium half-sun-exposed leaves, and multicellular trichomes on the lower shade leaves [24,25]. This leaf heterogeneity likely represents a specific adaptation to the environment [25,26]. However, there is limited information available on how *E. angustifolia* leaves respond to environmental conditions. In this study, we conducted submicroscopic structure observations of *E. angustifolia* leaves, performed in situ water-collection experiments, measured photosynthesis rates, and analyzed reflection spectra in order to determine their ability to collect water and reflect sunlight with their specifically constructed trichomes. This study provides the first insights into a novel mechanism employed by *E. angustifolia* leaves to acquire water and modulate light reflection, significantly contributing to our understanding of how plants with large and medium-sized leaves respond to environmental stresses in arid regions.

## 2. Materials and Methods

### 2.1. Sample Collection and Characterization

In July 2021, fresh adult *E. angustifolia* leaves were collected from the southern edge of the Mu Us Sandy Land (37°95′70″ N, 106°42′47″ E) in China. Healthy leaves were cut into 2 × 2 mm pieces, which were gold-spayed using an ion beam sputtering system (Hitachi E-1045, Tokyo, Japan) with a thickness of 20 nm. The microstructure of the leaf surface was observed with a field-emission scanning electron microscope (ESEM; Hitachi S-4800, Japan). The maximum magnification of the SEM was 650,000 times, the acceleration voltage was 0.530 kV; the resolution was 2.2 nm at 1 kV and 1.0 nm at 15 kV. The adaxial and abaxial leaf surfaces trichome size and morphology conducted SEM image comparison through MATLAB (Version R2022a.MathWorks, Inc., Natick, MA, USA) scripts to compute the trichome similarity.

### 2.2. Determination of Leaf Surface Wettability

A 1 μL water droplet was applied to the leaf surface, and the contact angle (CA) was measured using a Dataphysics OCA20 CA device (Dataphysics Inc., Filderstadt, Germany) with five replicates for each leaf within 30 s [27]. The measurement was conducted at room temperature (25.00 ± 1.00 °C), and the relative humidity was kept at 80%.

### 2.3. Water-Collection Experiment

The trichomes were carefully fixed on a glass slide, and purified water (Milli-Q Reference, Inc., Bedford, MA, USA) was added to an ultrasonic humidifier (Yadou YC-D204, Shanghai, China) to generate fog droplets to the leaf trichomes. Water collection of the trichomes was examined under the saturation status at a flow rate of 20–30 mm s^−1^ according to Ju [8]. The trichome region was imaged using a CCD color camera (Moticam Pro 252A; Motic Instruments Inc., Richmond, BC, Canada) attached to a compound microscope (BA410; Motic) at a magnification of ×10.

### 2.4. Photosynthesis Determination

A portable photosynthesis system (LI-6400; LI-COR, Inc., Lincoln, NE, USA) was used to measure the photosynthetic rate, intercellular CO_2_ concentration, stomatal conductance, and transpiration rate of the leaves according to Lawrence [28]. In July 2021, the southern edge (37°95′70″ N, 106°42′47″ E) of the Mu Us Sandy Land in China was utilized. Experimental results were tested statistically via paired two-tailed Student’s *t*-test and expressed as mean ± SEM by OriginPro (Version 2023. OriginLab Corporation, Northampton, MA, USA). Data were considered statistically significant when *p* < 0.05.

### 2.5. Spectral Measurement and Statistical Analysis

The spectra of the front and back surfaces of *E. angustifolia* leaves were measured using a USB4000 miniature fiber optic spectrometer (Ocean Optics, Inc., Dunedin, FL, USA), with a spectral range of 200–850 nm. Before measurement, a standard whiteboard was used for calibration, and the viewing angle was 25°. Three replicates were performed for each sample, and the mean value was taken as the spectral reflectance of this sample. R v3.6.0 and Origin v20.0 were used to analyze and plot the different data.

## 3. Results

### 3.1. Leaf Surface Characteristics

*E. angustifolia* has lanceolate leaves (Figure 1a), the adaxial and abaxial surfaces of which are covered with shiny, silvery-white trichomes. There are fewer trichomes (with a coverage of <37%) on the adaxial leaf surface (Figure 1b), while trichomes cover the entire abaxial leaf surface (Figure 1c). No significant difference in the trichome size and morphology was observed between the adaxial and abaxial leaf surfaces. There are many stomata on the epidermis of the abaxial leaf surface (Figure 2e)

### 3.2. Structure and Characteristics of Trichomes

A scanning electron microscope (SEM) was used to further observe the individual trichomes on the *E. angustifolia* leaf surface. The trichome is in the shape of an umbrella-shaped ratchet, harboring a plurality of wedge-shaped grooved two-dimensional (2D) umbrella surfaces composed of multiple tapered conical spines (Figure 2a). There are micropapillae on the apex of the umbrella surface, and the ratchet structure is composed of multiple one-dimensional (1D) cones extending radially from the periphery of the umbrella surface (Figure 2b). The umbrella-shaped ratchet trichomes are hollow (Figure 2c), and the micropapillae on the apex of a single trichome have an apex angle (α) of 115.30 ± 9.30° (*n* = 20), a height (h) of 30.00 ± 9.10 μm (*n* = 25), a basal width of 58.20 ± 9.30 μm (*n* = 24), and h/w of 0.50 ± 0.20 (*n* = 24) (Figure 2d). A single conical spine has a length (L) of 186.27 ± 20.20 μm (*n* = 40), and a single conical spine has an angle (*θ*_1_) of 7.46 ± 1.72° (*n* = 40); the angle between two adjacent conical spines (*θ*_2_) is 15.48 ± 2.64° (*n* = 45) (Figure 2f). The above findings indicate that a single umbrella-shaped ratchet trichome has both 1D and 2D structures.

### 3.3. Leaf Wettability

The front and back surfaces of *E. angustifolia* leaves are micro-/nano-level heterogeneous rough surfaces, mainly composed of a number of umbrella-shaped ratchet trichomes. The wettability of the front and back leaf surfaces was measured. The contact angle of the adaxial leaf surface was 117.40 ± 5.40° (*n* = 8) (Figure 3a), and that of the abaxial leaf surface was 138.20 ± 3.70° (*n* = 8) (Figure 3b), suggesting that both leaf surfaces are hydrophobic interfaces with self-cleaning properties.

### 3.4. Multifunctional Hierarchical Water-Collection Mechanism of Trichomes

The water-collection experiment was carried out on the trichomes of the *E. angustifolia* leaf surface, and the results showed that these trichomes could capture and directionally transport water droplets (Figure 4, Figure 5 and Figure 6). There were many stomata (Figure 2e and Figure 7a) on the leaf surface. Water droplets were captured by a single conical spine at the edge of the ratchet trichome (Figure 4b and Figure 7b) and transported from the apex to the base of the conical spine through the Laplace pressure difference. Two adjacent conical spines formed an open “wedge”-shaped capillary gap for water collection (Figure 4c). The aggregated water was transported to the apex of the umbrella handle through the wedge-shaped grooves on the abaxial surface of the umbrella-shaped trichomes (Figure 7c,d). Then, the water flowed from the apex to the base of the umbrella handle, from where it flowed into the stomata on the leaf surface, entering the leaf interior (Figure 7e), and finally, the airborne water captured by the trichomes was absorbed by the leaf. Large droplets were aggregated by multiple conical spines onto the leaf surface, where they were absorbed by the stomata. Therefore, the umbrella-shaped ratchet trichomes on the *E. angustifolia* leaf surface harbor a multifunctional hierarchical water-collection mechanism (Figure 8).

### 3.5. Response of Leaves to the Light

At noon (12:00–14:00) on a cloudless day, the adaxial surface of *E. angustifolia* leaves curled or turned over with the increasing sunlight intensity, exposing the abaxial leaf surface to the sunlight (Figure 9a). From morning to noon, the photosynthetic rate of *E. angustifolia* leaves changed intermittently: the stomata gradually closed, the transpiration rate gradually decreased, and the intercellular CO_2_ concentration fluctuated instantaneously (Figure 9b). The carbon dioxide concentration on the leaves was higher than in the environment, while the temperature, water, and relative humidity on the leaves were lower than in the environment (Figure 10). Both the front and back surfaces of *E. angustifolia* leaves could absorb blue (400–520 nm) and red lights (610–720 nm), while they highly reflected rays in the UV (<380 nm) and NIR regions (>700 nm) (Figure 9c). The umbrella-shaped ratchet trichomes on the *E. angustifolia* leaf surface greatly reduced the absorption of solar radiation and heat by the leaves (Figure 9d and Figure 11).

## 4. Discussion

### 4.1. Occult Precipitation Is an Important Source of Water in Arid Regions

In arid regions with little precipitation and strong sunlight radiation, the relative air humidity fluctuates between 30% and 90% with the changing temperature [29]. The natural precipitation and occult precipitation (e.g., nucleation droplets and fog droplets) are two different sources for plant water collection [30], of which natural precipitation is the main source [31]. Although the water amount of natural precipitation is large, the duration is often short, with a long dry interval between two precipitation peaks. Occult precipitation accounts for 13% of the total surface water sources as an important source of water for plants to survive during the intermittent period between natural precipitation [32]. Different from natural precipitation, the water amount of occult precipitation, in which water is adsorbed on the surface of plants, is small [33]. However, its duration is longer and more stable than that of natural precipitation, which is of great significance to the survival of plants and small animals in arid regions [34]. 

Natural *E. angustifolia* forests are widely distributed in arid and semi-arid regions, and the abaxial surface of *E. angustifolia* leaves is fully covered with umbrella-shaped ratchet trichomes. In this study, observing *E. angustifolia* leaves on cloudy and windless mornings, we found that the abaxial surface of the leaves was moist. This indicates that the abaxial surface of *E. angustifolia* leaves can intercept the occult precipitation from the air, which is an important mechanism for *E. angustifolia* to adapt to the arid environment.

### 4.2. Umbrella-Shaped Ratchet Trichomes on the E. angustifolia Leaf Surface Can Capture and Directionally Transport Water Droplets

The wetting process in solid materials exists widely in nature; the homogeneous rough surface is hydrophilic [35], while the non-homogeneous rough surface is hydrophobic [36]. Materials with anisotropic wetting characteristics can control the movement behavior of droplets in a specific direction, enabling directional droplet transport [37]. A droplet can be driven by factors related to asymmetric gradients [38,39], thermodynamics [40,41], and chemistry [42,43]. For example, droplets on the surface of asymmetric objects can be driven to move toward regions with a larger radius due to the Laplace pressure generated by the asymmetric gradients.

The phenomenon that water accumulates on the leaf surface and enters the leaf interior is called foliar water uptake (FWU), which can be facilitated with the assistance of stomata, cuticles [44], trichomes [45], or hydathodes [46]. Fog droplets usually harbor a smaller diameter (1–15 µm) compared to rain droplets (0.5–5 mm) [47]. On a nanostructure interface, a single microdroplet with a diameter of 2–4 μm gradually forms an approximately spherical droplet with a diameter of 4–6 μm [48,49]. A single umbrella-shaped ratchet trichome on the *E. angustifolia* leaf surface exhibited a 3D interface composed of anisotropic 1D and 2D structures (Figure 2a). The 1D ratchet structure was composed of a plurality of radial cones with an acute apex angle (*θ*_1_ = 7.46 ± 1.72°) and a micro-/nanostructure surface (Figure 2b,f). The conical structure [8,38,50] produced the Laplace pressure difference between two asymmetric gradients of the droplet. The capillary force (Figure 2b,f) generated by the angle between two adjacent conical spines (*θ*_2_ = 15.48 ± 2.64°) could further drive the droplets to the base of the spine. On the surface of 2D materials, the directional transport of liquids can be achieved through surface-energy gradients and Laplace pressure gradients. It has been reported that the water-collection ability is enhanced with the size of the patterned surface with a star-shape decrease from 1000 to 250 μm [51]; the patterned surface with a wedge-shaped array structure can also efficiently collect water [52]. The 2D umbrella surface structure of the ratchet trichomes is a patterned surface composed of multiple wedge-shaped groove structures (Figure 2a), which can also enhance the water-collection ability. In addition, since CO_2_ diffuses 10,000 times slower in water than in air [53], there is a strong selective pressure on land plant leaves to enhance water repellency. Trichomes on the leaf surface have a great influence on water repellency of the leaf (i.e., the extent to which droplets form on the leaf surface), as well as on the retention of droplets on the leaf [54]. The micropapillary structure on the apex of the trichomes (Figure 2b) increased the heterogeneous roughness of the leaf, making the leaf superhydrophobic (Figure 3).

The water-collection process of the 3D interface structure of the trichomes on the *E. angustifolia* leaf surface exhibited a hierarchical asymmetric effect. During the nucleation process, the droplets first randomly condensed on the spine surface (Figure 4b,c and Appendix A) and then converged toward the base of the spine, where the droplets aggregated to form larger droplets and moved from the semi-conical structure to the patterned surface with wedge-shaped grooves. Then, the larger droplets moved along the patterned surface (Figure 5b–f and Appendix A), enabling the directional transport of droplets from micro-trichome to macro-trichome (Figure 6b–f and Figure 7, Appendix A). However, the water-collection efficiency of trichomes, either a single trichome or multiple trichomes, has not been thoroughly investigated in this study, which needs to be further examined in the future. Compared with other water-collection systems, such as 1D fibers [8,9], highly irregular 2D surfaces [55], and 3D leaf–trichome hierarchical structures [27], each umbrella-shaped ratchet trichome on the *E. angustifolia* leaf surface is an independent microsystem with a 3D interface structure composed of 1D and 2D structures that can collect water hierarchically (Figure 8), enabling the capture of water droplets and the directional transport of water. In the natural environment, the apex of the trichomes on the abaxial surface of *E. angustifolia* leaves is facing the ground, like many umbrella-shaped containers hanging upside down. The water droplets in the air aggregate into larger droplets via the spines on the periphery of the umbrella-shaped structure and are transported to the umbrella-shaped container for temporary storage through the umbrella surface, which strengthens the collection and retention of water droplets and prevents the collected large droplets from falling on the ground and being lost.

### 4.3. Leaves Are the Main Plant Organ Responding to Solar Radiation

The spectrum of sunlight has a wide range, including UV (100–400 nm, 5%), visible (400–700 nm, 46%), and infrared rays (>700 nm, 49%), among which NIR lights (700–1100 nm) are the major source of heat on the Earth’s surface [56,57]. Plants mainly absorb blue (400–520 nm) and red lights (610–720 nm) during photosynthesis and are most sensitive to NIR radiation [58]. The heat generated after the absorption of NIR radiation can drive the leaves to transpire [59,60]. In high-temperature regions with strong sunlight radiation, the cooling mechanism of plants is usually at the expense of losing transpiration-related water on the leaf surface. In other plants, the pubescent trichomes on the leaf surface can increase reflectivity, preventing damage to plants from radiative overheating [10,61]. Therefore, plant cooling in high-temperature regions is more challenging than heating. 

The surfaces of plant leaves are rarely completely flat, and their epidermal cells show a convex curvature (Figure 2c), resulting in slightly rounded, rounded, papillary, or even sharply conical shapes. It has been reported that the light-collection performance of plants depends on the morphology of the microstructure array on the surface of plant tissues [62]. Specifically, the leaves or petals of plants can combine the micro-/nanostructures of epidermal cells to reduce the reflection loss and redirect incident photons, eventually inducing the light-focusing effect. For example, the most common conical cell structure can reduce the specular reflection on the petal surface, which is attributed to the multiple reflections between the micropapillae on the apex of conical cells. The authors of designed a “silicon pyramid” array with a height of 20 μm and a vertex spacing of 4 μm on the surface of a solar panel [63]. Compared with the solar panel without the array, the temperature of the solar panel with the silicon pyramid array can be reduced by 17.6 K, and it also shows superhydrophobicity and self-cleaning properties. The leaf epidermis acts as the first layer of the optical boundary and plays a key role in controlling the entry of light into the leaf. Some studies have pointed out that leaf surface trichomes can reduce the damage to the proteins of mesophyll cells caused by incident UV radiation [64]. Other studies have pointed out that the density of trichomes on the *E. angustifolia* leaf surface has an impact on the reflectance index of the visible lights and infrared lights. Upon dehairing (trichomes), considerable decreases in the visible reflectance (400–700 nm) were accompanied by slight decreases in the infrared reflectance (700–1000 nm) [26]. In this study, the *E. angustifolia* leaves showed a strong reflection in the UV and NIR regions (Figure 9c). Studies have shown that the height (h), diameter (w), and radius (r) of the micropapillae on the apex of the conical cells on the petal surface can all affect the absorption of visible and UV [65]. The greater the micropapillary height (20.00 ± 4.00 to 79.90 ± 8.70 µm), the stronger the UV-light absorption; the larger the micropapillary apex angle (20° to 118°), the stronger the UV-light reflection [66]. There are micropapillae on the apex of the umbrella-shaped ratchet trichomes (Figure 1d,e), with a height of 30.00 ± 9.10 μm, a basal diameter of 58.20 ± 9.30 μm, an aspect ratio of 0.3–0.7, and an apex angle of 115.30 ± 9.30° (Figure 2d). Therefore, the micropapillae on the apex of the umbrella-shaped trichomes can reflect UV, avoiding damage to proteins of mesophyll cells from UV and assisting the visits of pollinators [67]. In contrast to reflectance at shorter wavelengths, the MIR reflectance spectra cuticular development is the main driving factor [68]. The multiple spines on the periphery of the ratchet trichome may also diffract lights, while the trichome can transmit red and blue lights and reflect NIR (Figure 11). It is possible that the air gap of the hollow structure of the trichomes can improve the directional reflectivity of NIR lights since the hollow structure provides a second planar interface, from which light can be specularly reflected. Moreover, the wedge-shaped structure patterning on the surface of the umbrella-shaped trichomes produces a combined optical response that is highly directional, which is also the key to light reflection and needs to be further validated in the future. In summary, the umbrella-shaped ratchet trichomes on the *E. angustifolia* leaf surface have the ability to filter sunlight.

## 5. Conclusions

In this study, we first reported that the trichomes on the leaf surface of *E. angustifolia*, a sandy sub-tree species that lives in arid and semi-arid regions, have excellent structural characteristics absorbing occult precipitation with a special umbrella-shaped ratchet structure, and the peripheral conical structure can effectively capture water droplets in the air. The radial cones can capture water droplets, which are transported directionally through the wedge-shaped groove structure of the umbrella surface to the central point of the umbrella handle and aggregate into larger droplets. These larger droplets eventually reach the base of the umbrella handle and enter the leaf interior through the stomata. In addition, trichomes also harbor a special structure that can reflect sunlight. At noon on a cloudless day, the leaf curls or turns over; thus, the umbrella-shaped ratchet trichomes on the abaxial surface of the leaf are exposed to the sunlight. The trichomes can selectively transmit blue and red lights and reflect the UV and NIR lights, reducing leaf temperature. This study is of significance for understanding how plants with medium-sized leaves respond to environmental stresses in arid regions, providing a valuable reference for bionic development and research in atmospheric water harvesting, seawater desalination, energy management, microfluidic control, and daytime radiative cooling.

## Figures and Tables

**Figure 1 biology-12-01024-f001:**
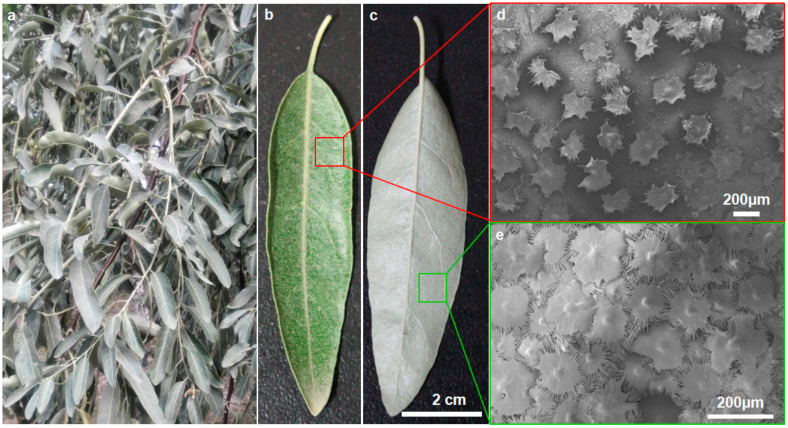
Morphological characteristics of *E. angustifolia* leaves. Medium-sized *E. angustifolia* leaves are lanceolate. (**a**) Wild *E. angustifolia*; (**b**) front surface of the leaf with sparse trichomes in dark green color; (**c**) back surface of the leaves with dense trichomes in white-gray color; (**d**) SEM of leaf front; (**e**) SEM of the back of the leaf.

**Figure 2 biology-12-01024-f002:**
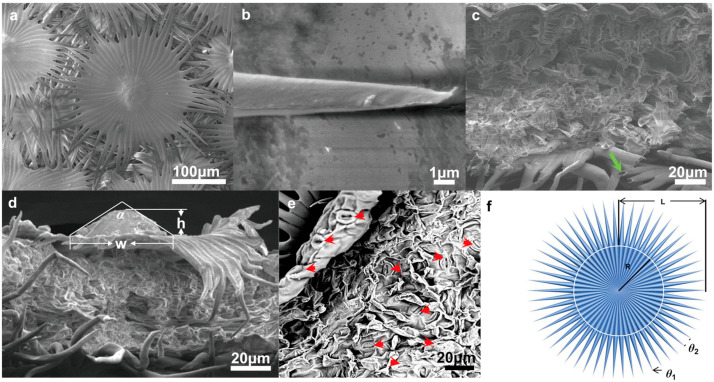
Submicroscopic structure of *E. angustifolia* leaves. An umbrella-shaped ratchet trichome of *E. angustifolia* leaves is composed of one-dimensional cone thorn and two-dimensional umbrella surface. (**a**) SEM image of umbrella ratchet trichome on leaf surface; (**b**) SEM image of trichome ratchet fiber; (**c**) SEM cross-section of *E. angustifolia* leaves, hollow structure of leaf trichomes (green solid arrow); (**d**) SEM of the structure of microemulsion on the top of a single trichome; (**e**) SEM image of the outer and inner layer structure on the back of the leaf with many stomata (red solid arrows); (**f**) an umbrella ratchet structure model of single trichome on the leaf surface of *E. angustifolia* composed of one-dimensional cone thorn and two-dimensional umbrella surface.

**Figure 3 biology-12-01024-f003:**
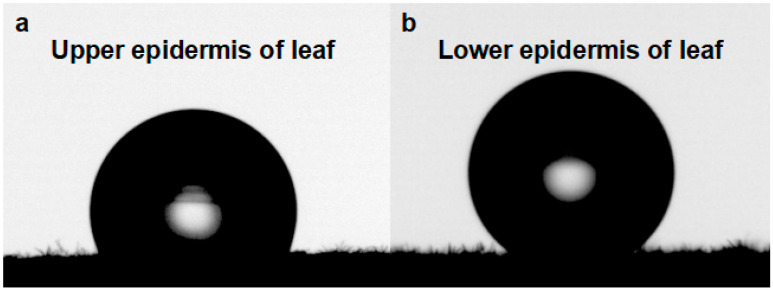
Leaf surface wettability of *E. angustifolia*. (**a**) The wettability of the epidermis on the leaves; (**b**) the wettability of the leaf lower epidermis.

**Figure 4 biology-12-01024-f004:**
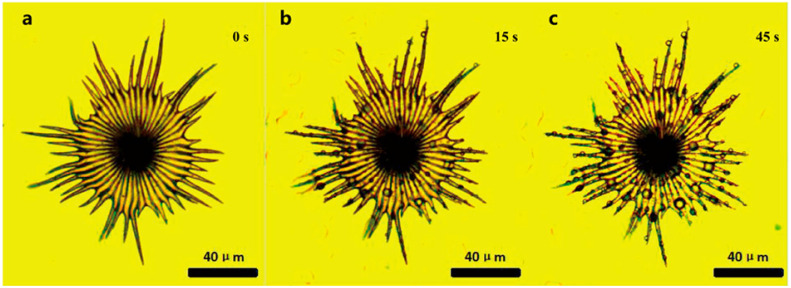
Water-collecting process of single trichome on the leaf surface of *E. angustifolia* (**a**→**b**→**c**). (**a**) The water-collecting of a single trichome at 0 s; (**b**) The water-collecting of a single trichome at 15 s; (**c**) The water-collecting of a single trichome at 45 s.

**Figure 5 biology-12-01024-f005:**
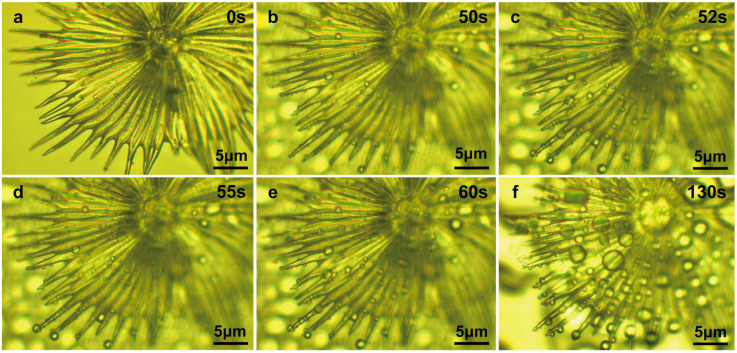
Water-collecting process of a single trichome with local magnification on the leaf surface of *E. angustifolia* (**a**→**b**→**c**→**d**→**e**→**f**).(**a**) The water-collecting of a single trichome with local magnification at 0 s; (**b**) The water-collecting of a single trichome with local magnification at 50 s; (**c**) The water-collecting of a single trichome with local magnification at 52 s; (**d**) The water-collecting of a single trichome with local magnification at 55 s; (**e**) The water-collecting of a single trichome with local magnification at 60 s; (**f**) The water-collecting of a single trichome with local magnification at 130 s.

**Figure 6 biology-12-01024-f006:**
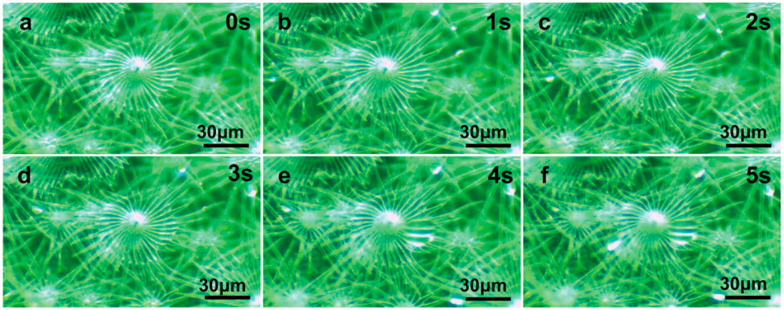
Water-collecting process of multiple trichomes on the leaf surface of *E. angustifolia* (**a**→**b**→**c**→**d**→**e**→**f**).(**a**) The water-collecting of multiple trichomes at 0 s; (**b**) The water-collecting of multiple trichomes at 1 s; (**c**) The water-collecting of multiple trichomes at 2 s; (**d**) The water-collecting of multiple trichomes at 3 s; (**e**) The water-collecting of multiple trichomes at 4 s; (**f**) The water-collecting ofmultiple trichomes at 5 s.

**Figure 7 biology-12-01024-f007:**
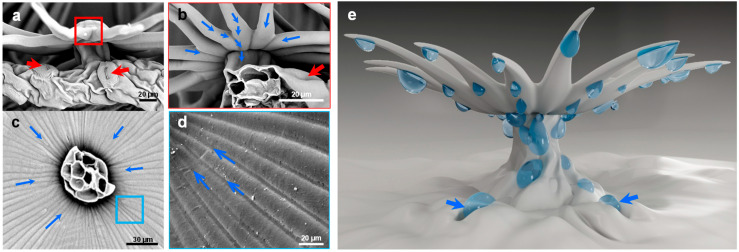
Water-collecting process of single trichome on leaf surface of *E. angustifolia*. (**a**) Side view of single umbrella ratchet trichome, in which the red arrow indicates the stomata on the leaf surface; (**b**) enlarged view of the trichome root of umbrella ratchet, in which the blue arrow represents the moving direction of droplets and the red arrow indicates the pores on the leaf surface; (**c**) the blue arrow in the figure represents the moving direction of the droplet; (**d**) the umbrella ratchet trichomes the groove structure on the back of the umbrella surface, and the blue arrow in the figure represents the moving direction of the droplet; (**e**) water-collection pattern of single trichome, in which the blue arrow indicates that the droplet enters the stomata on the leaf surface.

**Figure 8 biology-12-01024-f008:**
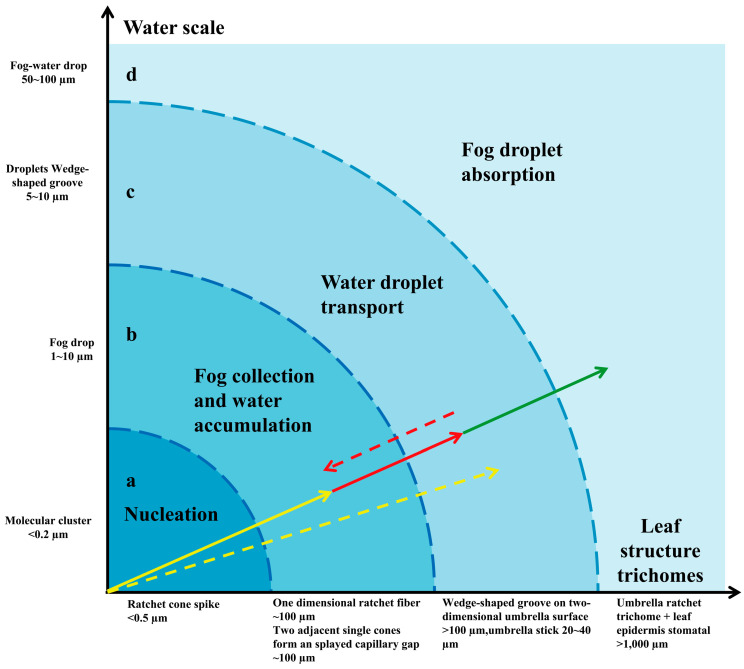
Multifunctional layered water-collecting mechanism of umbrella ratchet trichomes on leaf surface. The concentric shaded area shows the functional mapping between the size trichome of the structure on the umbrella ratchet trichome and the size trichome of water a–d. The arrow shows that the trichome is a catchment system composed of grading mechanisms, which together effectively capture all available water. a. Nucleation improves overall surface properties by providing a layer of water. The overall wettability is improved, providing more favorable nucleation, fog capture (yellow solid arrow), and water transport (yellow dotted arrow). b. Fog collection (one-dimensional cone prick fiber) and water accumulation (two adjacent single cones form an open octagonal capillary gap) gather large droplets (solid red arrow). c. The water droplets are transported on the back of the two-dimensional umbrella trichome (wedge-shaped groove), flow from the top of the umbrella handle to the base of the umbrella handle, enter the leaf surface pores, and prepare the next group of water droplets for the one-dimensional fiber cone to pierce the umbrella surface of the two-dimensional wedge-shaped groove (red dashed arrow). d. Large droplets are absorbed by multiple fiber cones (umbrella ratchet trichome clusters) and leaf surfaces (pores) (solid green arrows).

**Figure 9 biology-12-01024-f009:**
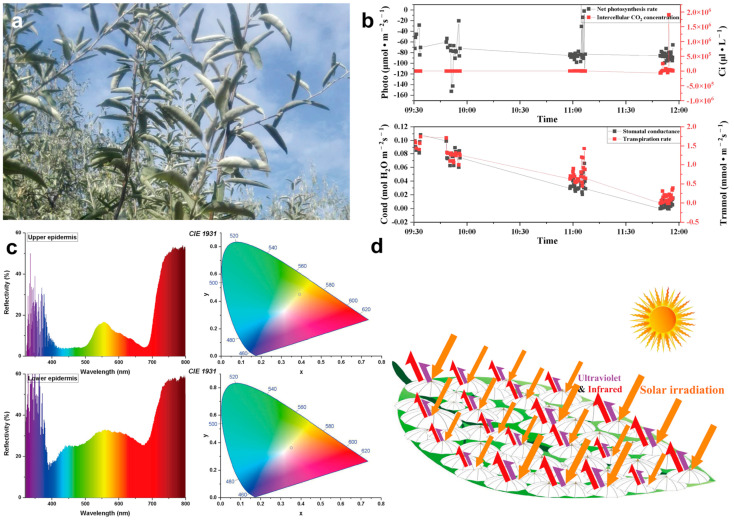
Response diagram of *E. angustifolia* leaves to light. When the leaves of *E. angustifolia* respond to strong sunlight, they reduce due to photosynthesis and irradiate the back of the leaves fully covered with trichomes to the sunlight so as to reduce the absorption of sunlight radiation heat by the leaves. (**a**) At noon, the leaves of *E. angustifolia* curl or turn over, exposing the back of the leaves to the sun; (**b**) changes in leaf photosynthetic rate (Net photosynthesis rate), intercellular CO_2_ concentration, stomatal conductance, and transpiration rate of *E. angustifolia* leaves; (**c**) reflection spectra of the front and back of *E. angustifolia* leaves; (**d**) pattern diagram of light response of umbrella ratchet trichomes on the leaf surface of *E. angustifolia*.

**Figure 10 biology-12-01024-f010:**
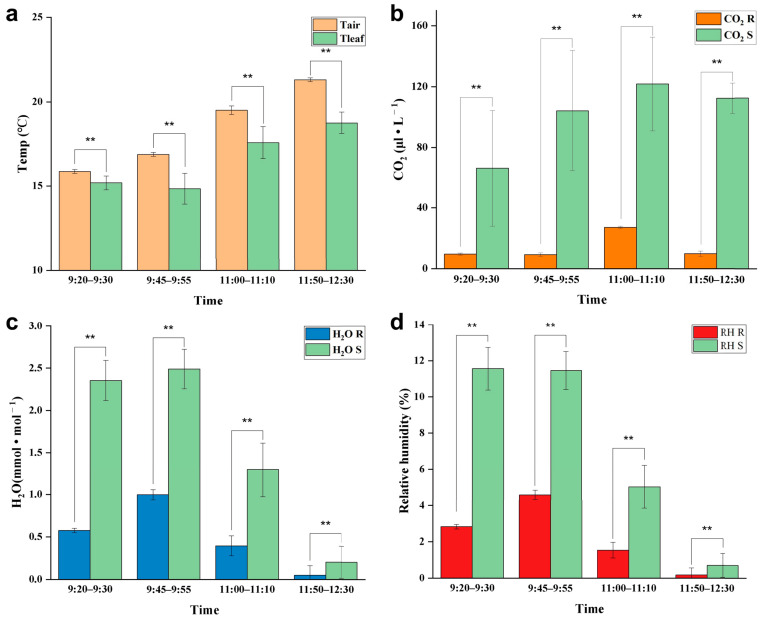
Response of *E. angustifolia* leaves to environment. (**a**) Change in temperature on the leaves with time (Tleaf is the temperature on the leaves; Tair is the temperature in the environment); (**b**) change in CO_2_ on the leaves with time (CO_2_ S is the CO_2_ on the leaves; CO_2_ R is the CO_2_ in the environment); (**c**) change in H_2_O on the leaves with time (H_2_O S is the CO_2_ on the leaves; H_2_O R is the H_2_O in the environment); (**d**) change in relative humidity on the leaves with time (RH S is the relative humidity on the leaves; RH R is the relative humidity in the environment). *p* values were determined by Student’s *t*-test (** *p* < 0.01).

**Figure 11 biology-12-01024-f011:**
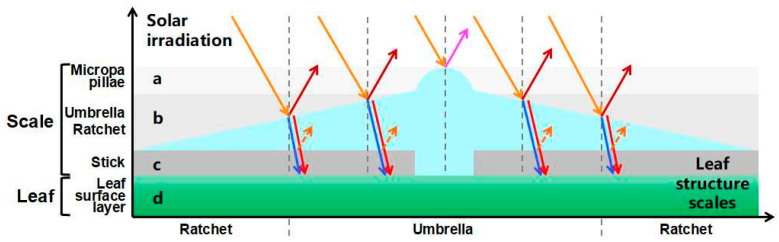
Screening of umbrella ratchet trichomes on leaf surface and mechanism of sunlight. The different shaded areas show the selection range a–c of different parts on the umbrella ratchet trichome for sunlight. All the arrows show the sun’s rays on the trichome. a. The microemulsion on the top of the umbrella ratchet trichome reflects the ultraviolet rays (solid purple arrow). b. The umbrella surface part of the umbrella like ratchet trichome reflects the near-infrared ray (solid dark red arrow), and the blue light (blue dotted arrow) and red light (red dotted arrow) reach the blade surface through the umbrella surface part. c. Gap between umbrella-shaped ratchet trichomes and the leaf surface. d. Leaf surface. The gray vertical dotted line is the normal line.

## Data Availability

This study did not generate a datasets.

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
