# Peer review of "The Mechanism by Which Umbrella-Shaped Ratchet Trichomes on the *Elaeagnus angustifolia* Leaf Surface Collect Water and Reflect Light"

_biology, 2023, doi:10.3390/biology12071024_

Round 1

Reviewer 1 Report

This is a nice paper with detailed analysis, but, as quite common these days, written as if completely new, etc. At first I had some difficulties figuring out where the name ratchet umbrella comes from.  In other studies such structures are called peltate,  stellate or stellar trichomes; and they occur for example in Olives (Bacelar, E. A., Correia, C. M., Moutinho-Pereira, J. M., Gonçalves, B. C., Lopes, J. I., & Torres-Pereira, J. M. (2004). Sclerophylly and leaf anatomical traits of five field-grown olive cultivars growing under drought conditions. Tree physiology, 24(2), 233-239. / ) the vernacular name of Eleagnus angustifolia  is wild olive indeed. Such types of trichomes occur in a variety of other plants;  It is necessary to add this type of information.

Other information that needs to be added is geographical information, details on the rainfall in these areas, the natural distribution of Eleagnus angustifolia.

The overall merit was rated average, because of these shortcomings.

Furthermore, the language needs serious editing (unfinished sentences,...

Author Response

Dear Editors and Reviewers:

Thank you for your letter and for the reviewers’ comments on our manuscript entitled “The mechanism of umbrella-shaped ratchet scales on the Elaeagnus angustifolia leaf surface to collect water and reflect light ” (ID: 2488989). These comments are all valuable and very helpful for revising and improving our paper, as well as the important guiding significance to our researches. We have carefully read the comments and questions, and made the following revisions (highlighting in red). We hope these changes will address the Editor and reviewers’ concerns.

We thank you for your willingness to consider the publication of this manuscript and your kind assistance with the reviewing process. We look forward to receiving your final decision for the revision.

Point 1: This is a nice paper with detailed analysis, but, as quite common these days, written as if completely new, etc. At first I had some difficulties figuring out where the name ratchet umbrella comes from.  In other studies such structures are called peltate,  stellate or stellar trichomes; and they occur for example in Olives (Bacelar, E. A., Correia, C. M., Moutinho-Pereira, J. M., Gonçalves, B. C., Lopes, J. I., & Torres-Pereira, J. M. (2004). Sclerophylly and leaf anatomical traits of five field-grown olive cultivars growing under drought conditions. Tree physiology, 24(2), 233-239. / ) the vernacular name of Eleagnus angustifolia  is wild olive indeed. Such types of trichomes occur in a variety of other plants;  It is necessary to add this type of information.

Response 1: We gratefully appreciate for your valuable comment. I have made comprehensive modifications in the introduction.The information is as follows:

The leaves of E. angustifolia are medium-sized and lanceolate in shape. They possess trichomes on both the upper and lower surfaces, exhibiting a silver-white color [23,24]. Upper sun leaves are smaller, more slender, and thicker than the lower, half-exposed shade leaves [25]. The foliage of a single E. angustifolia plant displays a gradient from silver-white to dark green, with peltate trichomes on the upper sun-exposed leaves, pedestalled trichomes on the medium half-sun-exposed leaves, and multicellular trichomes on the lower shade leaves [24,25]. This leaf heterogeneity likely represents a specific adaptation to the environment [25,26].

Point 2: Other information that needs to be added is geographical information, details on the rainfall in these areas, the natural distribution of Eleagnus angustifolia.

Response 2: Thank you so much for your careful check.We have added the following content to the introduction: Elaeagnus angustifolia L. (Elaeagnaceae) is a deciduous tree or large multilayer shrub with small size and spines. It is widely distributed in arid and semi-arid areas [11]. This species is commonly known as "Russian olives" or "oleaster" [12] and is native to southern Europe, Central Asia, and the western Himalayas [13]. During the early 20th century, it was introduced from Eurasia to Canada, the United States, the Mediterranean coast, Southern Russia, Iran, and India [14]. The fruit of E. angustifolia is highly nutritious, containing proteins, sugars, vitamins, and minerals, making it a valuable natural food and medicinal herb [15]. The climate conditions suitable for E. angustifolia are mainly found in cold and dry warm and mid-temperate zones in winter, as well as in arid and semi-arid regions between 30°N and 50°N in China [16]. E. angustifolia thrives in these areas, where the annual rainfall does not exceed 300 mm [17-20].  

Point 3: The overall merit was rated average, because of these shortcomings.

Response 3: Thank for your comments.

Point 4: Furthermore, the language needs serious editing (unfinished sentences,...

Response 4: We followed the reviewer’s advice. The manuscript has been carefully reviewed by an experienced editor whose first language is English and who specializes in editing papers written by scientists whose native language is not English. Certificate of English Editing:

Reviewer 2 Report

Dear Authors, please see the comments regarding your manuscript below.

Abstract:

·        Although it is written well it lacks descriptive information pertaining to the materials and methods and results and mainly focuses on a discussion of the results. Please kindly revise and include the necessary information in an abstract.

Keywords: The species name should be in italics.

Introduction:

·        Lack of reference to literature. Please ensure that each sentence in the introduction and discussion contains a reference to literature.

·        Kindly elaborate on trichomes, their general function, types and benefits with more examples, especially within the genus/ family of Elaeagnus angustifolia.

·        Give a better description of Elaeagnus angustifolia.

·        Elaborate on the location of the regions where it occurs.

Materials and methods:

·        What developmental stages of leaves were collected? Usually, younger leaves possess an abundance of scales and this decreases as the leaf matures.

·        If the leaves were collected fresh, what types of fixations were conducted before the observation with SEM? If leaves were not fixated, then an ESEM should have been used. Furthermore, the ultrastructure refers to using TEM and not SEM. Please correct.

·        For the water collection experiment, how were images captured?

Results

·        Please refer to the classification of trichomes and obtain the trichome type of this scale.

·        The images are of excellent quality.

Discussion

·        Well written with reference to literature.

Minor changes to English, grammar and formatting required. 

Author Response

Dear Editors and Reviewers:

Thank you for your letter and for the reviewers’ comments on our manuscript entitled “The mechanism of umbrella-shaped ratchet scales on the Elaeagnus angustifolia leaf surface to collect water and reflect light ” (ID: 2488989). These comments are all valuable and very helpful for revising and improving our paper, as well as the important guiding significance to our researches. We have carefully read the comments and questions, and made the following revisions (highlighting in red). We hope these changes will address the Editor and reviewers’ concerns.

We thank you for your willingness to consider the publication of this manuscript and your kind assistance with the reviewing process. We look forward to receiving your final decision for the revision.

Point 1: Abstract:

  • Although it is written well it lacks descriptive information pertaining to the materials and methods and results and mainly focuses on a discussion of the results. Please kindly revise and include the necessary information in an abstract.

Response 1: We gratefully appreciate for your valuable comment. We have made modifications to the abstract:Leaves are essential for plants, enabling photosynthesis and transpiration. In arid regions, water availability limits plant growth. Some plants, like Elaeagnus angustifolia, a sandy subtree species widely distributed in arid and semi-arid regions, have unique leaf structures to reduce water loss and solar radiation. Here, we describe the leaves of Elaeagnus angustifolia L., with special functioning trichomes.Through leaf submicroscopic structure observation, in situ water collection experiments, photosynthesis measurements, and reflection spectrum analysis, we investigated E. angustifolia leaves, focusing on their functioning trichomes. These trichomes capture water vapor, reflect UV and NIR light, and possess a 3D interface structure composed of 1D and 2D structures. The 1D conical structure captures water droplets, which are then gathered by the radial conical structure and guided towards the stomata through wedge-shaped grooves on the 2D umbrella structure. The trichomes also reflect sunlight, with micropapillae reflecting UV light and the umbrella structure reflecting NIR light. These mechanisms reduce leaf temperature, respiration, and water transpiration, protecting against solar radiation damage. This study provides insights into water collection and light reflection mechanisms, revealing adaptive strategies of plants with large leaves in arid regions.

Point 2: Keywords: The species name should be in italics.

Response 2: Thank you so much for your careful check. Keywords: Elaeagnus angustifolia leaf; umbrella-shaped ratchet trichomes; water collection; light reflection

Point 3: Introduction:

  • Lack of reference to literature. Please ensure that each sentence in the introduction and discussion contains a reference to literature.
  • Kindly elaborate on trichomes, their general function, types and benefits with more examples, especially within the genus/ family of Elaeagnus angustifolia.
  • Give a better description of Elaeagnus angustifolia.
  • Elaborate on the location of the regions where it occurs.

Response 3: Thank you for your rigorous nice suggestion. Modified Introduction:In arid regions characterized by clear skies and intense sunlight [1-3], efficient water acquisition, water conservation, and reduced solar radiation and heat absorption are critical for the survival of plants [4-6]. Plant leaves, like other organs, experience severe drought stress [7]. Consequently, leaves have developed an ingenious and effective mechanism to absorb water and reflect light in response to environmental stress. For instance, Opuntia microdasys leaves in desert areas possess specialized spines and glandular trichomes that minimize solar radiation and water loss during gas exchange and transpiration. Moreover, these clustered needle-like spines enable the directional collection of airborne water droplets [8], which sustain cactus survival in arid environments for several days. Another example is Syntrichia caninervis, a perennial plant that thrives in arid regions. Its leaves feature long white "awns" at their tips, which efficiently collect and transport water from the air while also reflecting sunlight to minimize water evaporation. This adaptation allows S. caninervis to flourish in arid environments [9]. In semi-humid regions, the large palmate leaves of Populus tomentosa possess a non-wettability hollow white tomentose layer on their back surfaces, which reflects up to 55% of sunlight. This feature significantly reduces leaves’ heat absorption and shields them from intense solar radiation [10]. However, little is known about the drought-resistant strategies of plants with large and medium-sized leaves in arid and semi-arid regions.

Elaeagnus angustifolia L. (Elaeagnaceae) is a deciduous tree or large multilayer shrub with small size and spines. It is widely distributed in arid and semi-arid areas [11]. This species is commonly known as "Russian olives" or "oleaster" [12] and is native to southern Europe, Central Asia, and the western Himalayas [13]. During the early 20th century, it was introduced from Eurasia to Canada, the United States, the Mediterranean coast, Southern Russia, Iran, and India [14]. The fruit of E. angustifolia is highly nutritious, containing proteins, sugars, vitamins, and minerals, making it a valuable natural food and medicinal herb [15]. The climate conditions suitable for E. angustifolia are mainly found in cold and dry warm and mid-temperate zones in winter, as well as in arid and semi-arid regions between 30°N and 50°N in China [16]. E. angustifolia thrives in these areas, where the annual rainfall does not exceed 300 mm [17-20]. Due to its ability to withstand severe drought, high salinity, cold, and wind stress [21], E. angustifolia plays a crucial role in preserving ecosystem functionality in arid areas [22].

The leaves of E. angustifolia are medium-sized and lanceolate in shape. They possess trichomes on both the upper and lower surfaces, exhibiting a silver-white color [23,24]. Upper sun leaves are smaller, more slender, and thicker than the lower, half-exposed shade leaves [25]. The foliage of a single E. angustifolia plant displays a gradient from silver-white to dark green, with peltate trichomes on the upper sun-exposed leaves, pedestalled trichomes on the medium half-sun-exposed leaves, and multicellular trichomes on the lower shade leaves [24,25]. This leaf heterogeneity likely represents a specific adaptation to the environment [25,26]. However, there is limited information available on how E. angustifolia leaves respond to environmental conditions. In this study, we conducted submicroscopic structure observations of E. angustifolia leaves, performed in situ water collection experiments, measured photosynthesis rates, and analyzed reflection spectra in order to determine their ability to collect water and reflect sunlight with their specifically constructed trichomes. This study provides the first insights into a novel mechanism employed by E. angustifolia leaves to acquire water and modulate light reflection, significantly contributing to our understanding of how plants with large and medium-sized leaves respond to environmental stresses in arid regions.

Point 4: Materials and methods:

  • What developmental stages of leaves were collected? Usually, younger leaves possess an abundance of scales and this decreases as the leaf matures.
  • If the leaves were collected fresh, what types of fixations were conducted before the observation with SEM? If leaves were not fixated, then an ESEM should have been used. Furthermore, the ultrastructure refers to using TEM and not SEM. Please correct.
  • For the water collection experiment, how were images captured?

Response 4: Thank you for the above suggestions. Fresh adult E. angustifolia leaves were collected from the southern edge of the Mu Us Sandy Land (37°95′70″N, 106°42′47″E) in China.

The microstructure of the leaf surface was observed with a field-emission scanning electron microscope (ESEM; Hitachi S-4800, Japan).

The trichome region was imaged using a CCD color camera (Moticam Pro 252A; Motic, British Columbia, Canada) attached to a compound microscope (BA410; Motic) at a magnification of ×10.

Point 5: Results

  • Please refer to the classification of trichomes and obtain the trichome type of this scale.
  • The images are of excellent quality.

Response 5: Thank you for your rigorous comment. Change scale to trichomes.

Point 6: The overall merit was rated average, because of these shortcomings.

Response 6: Thank for your comments.

Point 7: Discussion

  • Well written with reference to literature.

Response 7: We thank the reviewer for reading our paper carefully and giving the above positive comments.

Reviewer 3 Report

Reviewer’s comments on manuscript biology-2488989

The authors have examined leaves of Elaeagnus angustifolia L. in order to reveal would the specifically constructed trichomes on its surface can collect water and reflect light. The general idea of the paper is well formulated, since there is a lack of a coherent description of how structural characteristic of leaf surface of E. angustifolia contribute to environmental stresses in arid region. Using appropriate methods, the authors obtained valuable data and presented them in a good way. Results provide a comprehensive, detailed and precise description of the examined species. Figures are appropriate, informative, well organized and detailed.

The paper is an original contribution to the field because no detail data exist about function and structural characteristics of trichomes on leaves of E. angustifolia. In my opinion, this manuscript is acceptable for publication in this journal, after moderate revision.

My specific comments on the paper are listed below. I hope that the authors will find my comments useful and well-intentioned.

Specific comments:

Pages 1, 2, lines 32 – 53: The Introduction section should contain more information about the biology of the analyzed species, about the micromorphological characteristics of its leaves, so the readers can have a comprehensive picture of the material on which the analysis was made.

Page 2, lines 50 – 53: Authors should emphasize the aim of their work here instead of conclusion. For example: “We conduct a comprehensive study on lanceolate medium-sized leaf of Elaeagnus angustifolia L. in order to determine its ability to collect water and reflect sunlight with its specifically constructed trichomes.

Page 2, line 53: Space missing before word “with”.

Page 3, line 92: The authors use the term "scale" for the trichomes on the leaf surface of the analysed species. The term trichomes are more appropriate. Please make the changes throughout the text.

Page 3, line 112: The sentence about stomata is redundant in the description of trichomes. Move it to a paragraph in the general description of the leaf micromorphology of analysed species.

Page 4, lines 121, 122: It's not clear what the green arrow is pointing to.

Page 11, line 322: Part of sentence “and petals” is redundant. Figure 2 shows only the leaf detail.

Author Response

Dear Editors and Reviewers:

Thank you for your letter and for the reviewers’ comments on our manuscript entitled “The mechanism of umbrella-shaped ratchet scales on the Elaeagnus angustifolia leaf surface to collect water and reflect light ” (ID: 2488989). These comments are all valuable and very helpful for revising and improving our paper, as well as the important guiding significance to our researches. We have carefully read the comments and questions, and made the following revisions (highlighting in red). We hope these changes will address the Editor and reviewers’ concerns.

We thank you for your willingness to consider the publication of this manuscript and your kind assistance with the reviewing process. We look forward to receiving your final decision for the revision.

Point 1: Pages 1, 2, lines 32 – 53: The Introduction section should contain more information about the biology of the analyzed species, about the micromorphological characteristics of its leaves, so the readers can have a comprehensive picture of the material on which the analysis was made.

Response 1: Thank you for your rigorous nice suggestion. Modified Introduction: In arid regions characterized by clear skies and intense sunlight [1-3], efficient water acquisition, water conservation, and reduced solar radiation and heat absorption are critical for the survival of plants [4-6]. Plant leaves, like other organs, experience severe drought stress [7]. Consequently, leaves have developed an ingenious and effective mechanism to absorb water and reflect light in response to environmental stress. For instance, Opuntia microdasys leaves in desert areas possess specialized spines and glandular trichomes that minimize solar radiation and water loss during gas exchange and transpiration. Moreover, these clustered needle-like spines enable the directional collection of airborne water droplets [8], which sustain cactus survival in arid environments for several days. Another example is Syntrichia caninervis, a perennial plant that thrives in arid regions. Its leaves feature long white "awns" at their tips, which efficiently collect and transport water from the air while also reflecting sunlight to minimize water evaporation. This adaptation allows S. caninervis to flourish in arid environments [9]. In semi-humid regions, the large palmate leaves of Populus tomentosa possess a non-wettability hollow white tomentose layer on their back surfaces, which reflects up to 55% of sunlight. This feature significantly reduces leaves’ heat absorption and shields them from intense solar radiation [10]. However, little is known about the drought-resistant strategies of plants with large and medium-sized leaves in arid and semi-arid regions.

Elaeagnus angustifolia L. (Elaeagnaceae) is a deciduous tree or large multilayer shrub with small size and spines. It is widely distributed in arid and semi-arid areas [11]. This species is commonly known as "Russian olives" or "oleaster" [12] and is native to southern Europe, Central Asia, and the western Himalayas [13]. During the early 20th century, it was introduced from Eurasia to Canada, the United States, the Mediterranean coast, Southern Russia, Iran, and India [14]. The fruit of E. angustifolia is highly nutritious, containing proteins, sugars, vitamins, and minerals, making it a valuable natural food and medicinal herb [15]. The climate conditions suitable for E. angustifolia are mainly found in cold and dry warm and mid-temperate zones in winter, as well as in arid and semi-arid regions between 30°N and 50°N in China [16]. E. angustifolia thrives in these areas, where the annual rainfall does not exceed 300 mm [17-20]. Due to its ability to withstand severe drought, high salinity, cold, and wind stress [21], E. angustifolia plays a crucial role in preserving ecosystem functionality in arid areas [22].

The leaves of E. angustifolia are medium-sized and lanceolate in shape. They possess trichomes on both the upper and lower surfaces, exhibiting a silver-white color [23,24]. Upper sun leaves are smaller, more slender, and thicker than the lower, half-exposed shade leaves [25]. The foliage of a single E. angustifolia plant displays a gradient from silver-white to dark green, with peltate trichomes on the upper sun-exposed leaves, pedestalled trichomes on the medium half-sun-exposed leaves, and multicellular trichomes on the lower shade leaves [24,25]. This leaf heterogeneity likely represents a specific adaptation to the environment [25,26]. However, there is limited information available on how E. angustifolia leaves respond to environmental conditions. In this study, we conducted submicroscopic structure observations of E. angustifolia leaves, performed in situ water collection experiments, measured photosynthesis rates, and analyzed reflection spectra in order to determine their ability to collect water and reflect sunlight with their specifically constructed trichomes. This study provides the first insights into a novel mechanism employed by E. angustifolia leaves to acquire water and modulate light reflection, significantly contributing to our understanding of how plants with large and medium-sized leaves respond to environmental stresses in arid regions.

Point 2: Page 2, lines 50 – 53: Authors should emphasize the aim of their work here instead of conclusion. For example: “We conduct a comprehensive study on lanceolate medium-sized leaf of Elaeagnus angustifolia L. in order to determine its ability to collect water and reflect sunlight with its specifically constructed trichomes.”

Response 2: Thank you for pointing out this problem in manuscript. lines 50 – 53:  In this study, we conducted submicroscopic structure observations of E. angustifolia leaves, performed in situ water collection experiments, measured photosynthesis rates, and analyzed reflection spectra in order to determine their ability to collect water and reflect sunlight with their specifically constructed trichomes. This study provides the first insights into a novel mechanism employed by E. angustifolia leaves to acquire water and modulate light reflection, significantly contributing to our understanding of how plants with large and medium-sized leaves respond to environmental stresses in arid regions.

Point 3: Page 2, line 53: Space missing before word “with”.

Response 3: Thank you for pointing out this problem in manuscript. The paragraph has been modified.

Point 4:Page 3, line 92: The authors use the term "scale" for the trichomes on the leaf surface of the analysed species. The term trichomes are more appropriate. Please make the changes throughout the text.

Response 4: Thank you for the suggestions. The entire text has been modified.

Point 5: Page 3, line 112: The sentence about stomata is redundant in the description of trichomes. Move it to a paragraph in the general description of the leaf micromorphology of analysed species.

Response 5: Thank you so much for your careful check. It has been moved to 3.1

Point 6: Page 4, lines 121, 122: It's not clear what the green arrow is pointing to.

Response 6:Thank you for pointing out this problem in manuscript. It'shollow structure of the trichomes(green arrow). The blurry image has been replaced with a clear image.

Point 7: Page 11, line 322: Part of sentence “and petals” is redundant. Figure 2 shows only the leaf detail.

Response 7: Thank you so much for your careful check.Page 11, line 322: Part of sentence “and petals” deleted.

Round 2

Reviewer 1 Report

Congratulations on the improvement.  However before publication, a check on grammar should be performed

Italicize names Populus tomentosa an dSyntrichia caninervis

H2O and CO2: the two should be subscripts not same level 

spacing  example: ce of plant 375 tissues . Specifi

Author Response

Dear Editors and Reviewers:

Thank you for your letter and for the reviewers’ comments on our manuscript entitled “The Mechanism by Which Umbrella-Shaped Ratchet Trichomes on the Elaeagnus angustifolia Leaf Surface Collect Water and Reflect Light ” (ID: 2488989). These comments are all valuable and very helpful for revising and improving our paper, as well as the important guiding significance to our researches. We have carefully read the comments and questions, and made the following revisions (highlighting in red). We hope these changes will address the Editor and reviewers’ concerns.

We thank you for your willingness to consider the publication of this manuscript and your kind assistance with the reviewing process. We look forward to receiving your final decision for the revision.

Point 1: Congratulations on the improvement. However before publication, a check on grammar should be performed

Italicize names Populus tomentosa and Syntrichia caninervis

H2O and CO2: the two should be subscripts not same level 

Spacing example: ce of plant 375 tissues . Specifi

Response 1: Thank you for bringing this issue in the manuscript to my attention.

The names "Populus tomentosa" and "Syntrichia caninervis" have been appropriately italicized.

The subscripts for H2O and CO2 in the manuscript have been adjusted as per the suggestion.

The word spacing in the manuscript has been revised accordingly.

Reviewer 2 Report

Dear Authors, 

Thank you for your detailed changes, no further revisions are required. 

Minor editing of English is required. 

Author Response

Dear Editors and Reviewers:

Thank you for your letter and for the reviewers’ comments on our manuscript entitled “The Mechanism by Which Umbrella-Shaped Ratchet Trichomes on the Elaeagnus angustifolia Leaf Surface Collect Water and Reflect Light ” (ID: 2488989). These comments are all valuable and very helpful for revising and improving our paper, as well as the important guiding significance to our researches. We have carefully read the comments and questions, and made the following revisions (highlighting in red). We hope these changes will address the Editor and reviewers’ concerns.

We thank you for your willingness to consider the publication of this manuscript and your kind assistance with the reviewing process. We look forward to receiving your final decision for the revision.

Point 1: Comments and Suggestions for Authors

Dear Authors, 

Thank you for your detailed changes, no further revisions are required. 

Comments on the Quality of English Language

Minor editing of English is required. 

Response 1: We thank the reviewer for reading our paper carefully and giving the above positive comments.We followed the reviewer’s advice. The manuscript has been carefully reviewed by an experienced editor whose first language is English and who specializes in editing papers written by scientists whose native language is not English. Certificate of English Editing.
